# Developing Accurate Repetition Prediction Equations for Trained Older Adults with Osteopenia

**DOI:** 10.3390/sports12090233

**Published:** 2024-08-28

**Authors:** Rose Beia, Alfred Wassermann, Sebastian Raps, Jerry Mayhew, Michael Uder, Wolfgang Kemmler

**Affiliations:** 1Institute of Radiology, University Hospital Erlangen, Henkestrasse 91, 91052 Erlangen, Germany; rose.beia@fau.de (R.B.); michael.uder@uk-erlangen.de (M.U.); 2Department of Mathematics, University of Bayreuth, 95440 Bayreuth, Germany; alfred.wassermann@uni-bayreuth.de; 3Institute of Medical Physics, Friedrich-Alexander University Erlangen-Nürnberg, 91052 Erlangen, Germany; sebastian.raps@fau.de; 4Department of Health and Exercise Sciences, Truman State University, Kirksville, MO 63501, USA; jmayhew@truman.edu

**Keywords:** one repetition maximum, repetitions to fatigue, prediction equation, older people, osteopenia

## Abstract

The aim of this study was to evaluate prediction equations to estimate 1RM in different exercises in older men and women with osteopenia/osteoporosis. Forty well-trained older women and men (73 ± 8 years) with osteopenia/osteoporosis performed 1RM dynamic and isometric maximum strength tests on resistance devices. In addition, each participant performed repetitions-to-fatigue (RTF) in the 5–8RM, 9–12RM, and 13–16RM zones. After evaluating the predictive performance of available 1RM prediction equations from the literature, new prediction equations were developed for all seven exercises. One of the available equations that focus on postmenopausal women already acceptably predicted 1RM from RTF for all but one exercise. Nevertheless, new exercise-specific prediction equations based on a cubic polynomial most accurately predict 1RM from RTF in the 5–8 reps range with mean absolute differences between predicted and actual 1RM of 3.7 ± 3.7% (leg-press) to 6.9 ± 5.5% (leg flexion) that is roughly within the acceptable coefficient of variation. For some exercises, the inclusion of the isometric maximum strength tests slightly increases the prediction performance of the 5–8RM. In conclusion, the present prediction equation accurately estimates 1RM in trained, older women and men with osteopenia/osteoporosis. Further evaluation of this new equation is warranted to determine its applicability to different age groups and populations.

## 1. Introduction

Resistance exercise training (RT) is a key component of the guidelines and recommendations for the prevention and therapy of osteoporosis [1,2]. There is considerable evidence that dedicated resistance exercise protocols positively impacted the main determinants of fracture risk, i.e., decreased bone strength [3] and increased fall frequency [4]. However, an accurate specification of RT load prescription is essential for the effective and safe achievement of training aims. Proper specification of exercise intensity might be the most critical issue when designing RT protocols for adults. This is even more applicable for older adults (i.e., people 60 years and older) with increased fracture risk (i.e., osteoporosis).

On the other hand, strain magnitude [5,6] (i.e., mechanical strain induced by external loads) is a key factor for bone strength adaptation. Apart from bone strength, exercise intensity is usually the reference parameter for the progression and periodization of resistance exercise protocols. Thus, the relevance of accurately determining (high) resistance exercise intensity prescription is particularly important for people with osteopenia and osteoporosis. One repetition maximum tests (1RM) have been considered the “gold standard” of dynamic maximum strength evaluation [7]. Of importance, the frequently applied argument of increased risk of injuries and adverse effects with 1RM testing in older, or more precise vulnerable cohorts in general has been rejected by several studies [8,9,10]. Nevertheless, limited motivation to lift maximum loads, the requirement for high test effort, time constraints, and the aspect that the continuity of the training process is interrupted by dedicated 1RM tests, aggravate the frequent application of 1RM tests particularly in non-athletic RT. In contrast, 1RM prediction equations that use the load and number of repetitions completed to fatigue (RTF) to predict repetition maximum [11] might be a reliable and feasible alternative to 1RM testing. This rating is based on the rather unproblematic implementation of the RTF test into the training process. Considering the easy-to-perform demand that the number of reps prescribed by the training protocol has to be completed to fatigue indicates the potential of RTF tests for frequent prediction of adequate load with implication to individualization, specificity, progression and periodization of the training process.

Several prediction equations currently exist (e.g., [12,13,14,15,16,17]), although only a few focus on older adults [18,19]. Of note, two studies that determined the accuracy of present equations in untrained healthy older adults [20] or resistance-trained osteopenic postmenopausal women [18] reported differences of up to 20% between predicted and actual 1RM of the easy standardizable leg press exercise. In contrast, a more specific approach [18] based on data from the postmenopausal cohort mentioned above, revealed accurate prediction performance, i.e., differences between actual and predicted 1RM within the coefficient of variation for 1RM of the present cohort.

Thus, the primary aims of this study were to evaluate the predictive accuracy of current RTF equations and to develop an equation that more accurately predicts 1RM from RTF tests in trained older people with osteopenia and osteoporosis. Based on present findings [20,21], we hypothesize that the performance of a prediction equation specifically calculated for each test exercise will be significantly more accurate compared to present prediction equations or to a prediction equation based on data from a single exercise.

## 2. Methods

This project is part of an ongoing investigation aimed at gathering normative values for the proxomed compass 600 (Alzenau, Germany) resistance exercise device. The main objective of the current study was to develop prediction equations to accurately estimate one-repetition maximum performance for compass 600 devices for older men and women. This study is a cross-sectional investigation of older people with multiple years (≥five years) of experience in resistance exercise. The study protocol was approved by the ethics committee of the University of Erlangen (Ethikantrag 18-118-B and 287-18B). The project was conducted in complete adherence to the Helsinki Declaration “Ethical Principles for Medical Research Involving Human Subjects” [22]. After detailed information was provided, participants gave written informed consent before the study began.

### 2.1. Participants

Women and men 60 years and older from previous studies [23,24,25] and members of our university health sports club on bone health (Netzwerk Knochengesundheit e.V., Erlangen, Germany) were recruited through personal contact by the authors. Of importance, all participants continued exercise after the study ended and were transferred to a roughly similar fracture prevention exercise training that included components of weight bearing (e.g., aerobic dance, Polka, gaming) and periodized multiple-set resistance exercise on machines. 

The eligibility criteria for inclusion were (1) age 60 years and older. (2) Osteopenia (BMD <−1 SD T-Score) or osteoporosis BMD (<−2.5 SD T-Score) at the lumbar spine or total hip region of interest as determined by Dual-Energy X-ray Absorptiometry (Hologic QDR 4500a, Discovery-upgrade, Bedford, MI, USA) [26]. Assessments of BMD were performed at least 9 months before study start of the present project using standardized procedures and analyzed by a medical imaging expert of the Osteoporosis Research Center, Friedrich-Alexander-University of Erlangen-Nürnberg, Germany. (3) Continuous history of resistance exercise for at least five years, and (4) experience in 1RM test procedures and repetition to fatigue (RTF) tests. Excluded volunteers had: (1) non-treated hypertension, (2) musculoskeletal injuries that may prevent a reliable and safe assessment of the 1RM and RTF tests, and/or (3) people with an interruption of training continuity of more than 4 weeks during the last six months prior to the study. In summary, 40 participants (23 women and 17 men) were eligible and willing to participate in the study. 

### 2.2. Testing Procedures

All data were collected within an 8-month period during participants’ habitual training sessions. All participants were familiar with the testing devices. Experienced research assistants qualified by our specific university resistance training education consistently supervised 1RM and RTF tests performed on dedicated resistance training devices (Proxomed c600 series, Alzenau, Germany). Of importance, the Proxomed c600 series devices enable isometric maximum strength testing. Briefly, seven exercises (leg press, latissimus pulls, triceps dips, trunk extension, trunk flexion, leg extension, and leg flexion) were performed on the four devices illustrated in Figure 1. 

The maximum isometric strength, 1RM, and three RTF tests (5–8, 9–12, 13–16 repetitions) were completed during five 90-min training sessions at weekly intervals. Considering that exercises for related muscle groups or agonist/antagonist in immediate succession were not intended we do use a randomized test sequence. In the first session, the isometric and dynamic maximum strength tests were conducted, each with a 2–3 min rest between the isometric test and the 1RM procedure with a 5-min rest between the seven exercises. However, 1RM tests were not continued before participants confirmed full recovery. The standardized sequence of the 1RM test exercise was (1) 1RM leg press, (2) lat pulls, (3) trunk extension, (4) leg flexion (5) triceps dips (6) trunk flexion (7) leg extension. 

RTF in the 5–8RM repetition range was recorded in session 2, 9–12RM in session 3, and 13–16RM in session 4. Rest between RTF tests for the seven exercises average about 10 min. Apart from this approach, prior to the test, we asked participants if they felt completely regenerated. The sequence of the test exercises was similar to the 1RM test. 

After 5 min of aerobic warm-up with a cross-trainer once per test session, participants performed five reps at a non-repetition maximum [11] prior to the test on the dedicated device. 

### 2.3. Participants Positioning, Device Setting

All seven test exercises were performed in a statically and dynamically mode. With the exception of knee flexion/extension, the positioning in the static test was identical to the starting position of the dynamic test.

Leg press: Dynamic leg-press test was performed using a range of movement (ROM) between 90° and nearly extended knees at about 10–20° flexion (anatomical neutral standing position = 0°). The static testing was performed in 90° flexion. The individual device setting at 90° was documented for reproducibility using the device scaling. The backrest was in a 160° backwards inclined position. The feet were positioned so that the lower legs were horizontal to the floor. The foot plate was locked in device setting 2 (10% inclined backwards). 

Lat pulls: In order to ensure a correct sitting position during the test, the safety belt was fastened tightly. The seat height was adjusted according to body height to ensure a knee angle of approx. 90°. The test exercises were performed in an upright sitting posture with a predefined wide grip position. The isometric measurement position or starting point of the movement for dynamic testing was also adjusted to the body height so that the elbows were not fully extended at the beginning of the test exercise. The end of the movement of the dynamic test was when the handles reached shoulder height (acromion). 

Tricep dips: The sitting position for the tricep dips exercise was almost identical to the shoulder lat pulldown exercise with the difference that the upper body was leaning slightly forward. A 90° elbow angle was defined as the starting position. Similar to the previous tests, the end position for the dynamic test was also just before reaching full extension of the elbows, with the hands in a neutral grip position.

Trunk extension and -flexion: The same seat position was defined and recorded for trunk extension and -flexion. For flexion, the pad of the lever arm was adjusted according to body height at the height of the center of the collarbone. The pad was gripped with both arms. The isometric test position and starting angle of the dynamic tests for trunk flexion was always position 4 of the lever arm (almost upright position), and the endpoint was defined as elbows touching thighs. For trunk extension, isometric measurement position or dynamic test starting position was position 2 of the lever arm (chest is closer to the thighs). The pad of the lever arm was adapted to the body size and was positioned in the middle of the scapulae. The end position of the dynamic testing procedure was an upright sitting position with an extended spine.

Knee flexor/extensor: The device was adjusted so that the pivot point of the knee joint and the lever arm were exactly on the same level. The position for the isometric test was set at 15° in the knee joint for leg flexion and at 75° for the leg extension test. The dynamic measurement was performed in a ROM between 90° and (nearly) full extension, with the endpoint of the movement at 90° for leg flexion and about 0–10° for knee extension.

### 2.4. 1RM and RTF-Tests, Test Reproducibility

Based on the isometric maximum load (in N), participants were given a starting weight for 1RM attempts that was 15% lower than the isometric maximum. Loads were increased by a minimum of 1–2 kg until failure. Weight increases after successful attempts were conducted in close cooperation between participants and research assistants. 1RM was defined as the load of the last test that could be performed one time over the specified ROM. 1RM was usually realized within 4–5 trials. 

Load prescription for the RTF tests based on the foregoing results of the 1RM test. In detail, using the KLW-prediction equation [18] we calculate the load for the RTF tests (e.g., 82.5% 1RM for 5–8RM). In cases of RTF performance lower than 5 or higher than 16 reps, the test was repeated in the subsequent session. However, test results slightly above the prescribed repetitions range (e.g., nine reps within the 5–8RM) were considered for the subsequent 9–12RM range. Correspondingly the 5–8RM test was repeated in the next session. 

The reproducibility of the 1RM and 8RM tests in the present cohort was recorded in a random sample of 10 men and women 2–4 weeks prior to the project. To enhance test–retest (intra-rater) reliability, a screen in front of the weight blocks blinded the load. In summary, 1RM tests for all exercises were performed twice with a rest period of one week between the assessments. Rest periods between the 1RM tests average 15 min between the seven exercises. The identical test protocol was applied in both trials. This particularly includes the warm-up protocol, the sequence of exercises and the load increases up to failure. 8 RTF tests focus on the number of repetitions conducted with the identical load specified [18] for the 5–8 repetition range. In parallel to 1RM assessments 8RM tests for all exercises were conducted within one session with a rest of 10 min between the assessments. The repeated measure of 8 RTF was conducted one week after the first assessment. Again, the identical test procedures were applied. 

Scaling on each device allowed precise participant alignment to determine the identical position during the isometric maximum, 1RM, and RTF tests. All dynamic tests were conducted identically in a 2-s concentric–1-s isometric–2-s eccentric time under tension mode [18]. In summary, the following measurements were recorded for each of the seven exercises: isometric maximum, 1RM, 5–8RM, 9–12RM and 13–16RM. 

### 2.5. 1RM Prediction Equations Applied

Each 1RM test (leg press, latissimus pulls, triceps dips, trunk extension, trunk flexion, and leg extension-flexion) was estimated using seven RTF prediction models: Brzycki [12], Epley [13], Lander [14], Mayhew et al. [15], O’Conner et al. [16], Wathen [17] and Kemmler et al. [18] using the 5–8 RTF range. Table 1 summarizes the prediction equation of the authors.

### 2.6. Statistical Analysis

In order to test the prediction accuracy of the selected equations, 1RM values for each test (leg press, latissimus pulls, triceps dips, trunk extension, -flexion and leg extension, -flexion) were estimated using each prediction equation. Predicted mean and standard deviation (SD) for each 1RM test were compared with actual 1RM values using the Wilcoxon paired sample test or the paired t-test where applicable. Normal distribution was tested using Q-Q plots and the Shapiro–Wilks test. Sex differences (normally distributed) were addressed using an independent t-test. Statistical significance (2-tailed) was accepted at *p* ≤ 0.05.

## 3. Results

### 3.1. Participant Characteristics

Table 2 provides participant characteristics of the 17 men and 23 women. In summary, all participants had a long history of periodized RT in the range of 55–92.5% 1RM. Due to knee osteoarthritis, two participants were excluded from performing lower extremity exercise testing. Four participants with severe osteoporosis were excluded from 1RM tests of the back extensor and flexor exercises. A further four participants were unable to conduct all tests due to holidays, diseases or injuries unrelated to exercise. Finally, the 1RM of the two men was higher than the maximum possible load of the leg press. However, none of the participants withdrew from the study for personal reasons or discomfort. In summary, the number of participants with full data sets (isometric, 1RM, RTF with 5–8, 9–12 and 13–16 reps) varied between 32 (leg-press) and 36 (lat pulls). It is worth noting that no injuries or adverse effects were observed or reported after the 300 1RM and 800 RTF tests. 

### 3.2. Coefficients of Variation of 1RM and RTF-Tests

The coefficients of variation (CV) of the 1RM-tests ranged from 3.3 ± 1.3% for leg press, 3.3 ± 1.5 for lat pulls, 5.4 ± 2.1% for triceps dips, 3.6 ± 1.7% for trunk extension, 4.4 ± 2.1 for trunk flexion, 3.9 ± 1.8 for leg extension, and 4.6 ± 2.4% for leg flexion. The reproducibility of the 5–8RM repetition range slightly varies with the most reliable data for the leg press exercise (0.7 ± 0.7 reps) and the less reliable data for triceps dips 1.2 ± 1.1 reps, without differences between the genders.

### 3.3. Prediction Performance of 1RM Prediction Equations Applied to the Current Data Set

Table 3 provides M ± SD for predicted and actual 1RM values, significance levels for 1RM differences, and absolute values of the percentage difference between predicted 1RM and actual 1RM. All but one prediction equation [18] consistently overestimates 1RM. Differences between predicted and actual 1RM were most pronounced for the leg press exercise (5.4 to 12.9%). The less pronounced differences between actual and predicted 1RM average were trunk extension (4.1 ± 2.8%) and trunk flexion (8.1 ± 8.8%) provided by the KLW equation [18] developed on a comparable cohort of older people (Table 3).

### 3.4. Development of a New Prediction Equation

To estimate 1RM more accurately, new 1RM prediction equations were developed on the current data set for 8RM, 10RM, 15RM, ISOM, or combinations thereof. Using the method of least squares, a polynomial p(x) was determined from the data sets (1, w1), (r8, w8), (r10, w10), (r15, w15) that minimized the difference:|w1 − p(ri) × wi|, i ∈ {8, 10, 15}
where: wi is the load of measurement I and ri is the number of repetitions

The most accurate cubic curve (Figure 2) approximates all tests by:p(x) = 0.99157 + 0.01047492 · RTF + 0.00073974367 · RTF^2^ − 0.0000089288 · RTF^3^
i.e., regression coefficient: [9.9157 × 10^−1^, 1.04749226 × 10^−2^, 7.39743671 × 10^−4^, −8.92884730 × 10^−6^].

Isometric coefficient: 1.0002929780312122 (mean of 1RM/ISOM).

ISOM + r8 (Table 4 and Table 5) was calculated as the arithmetic mean between ISOM and r8, i.e., (ISOM + r8)/2.

During the first iteration, we computed the prediction equation for the leg press exercise and transferred the equation using the data from the other exercises (Table 4).

Table 4 displays absolute values of the difference between actual 1RM and predicted 1RM.

In summary, we observed high to acceptable prediction performance in the range of 3.70 ± 2.29 for lat pulls to 6.38 ± 5.71 for leg flexions. Nevertheless, compared with the KLW equation, significantly better prediction performance was only observed for the trunk flexion (5.28 ± 4.39 vs. 8.10 ± 8.82, *p* = 0.012) and leg press (3.72 ± 3.65 vs. 5.42 ± 3.30; *p* = 0.042) exercise for which a device-specific prediction was computed.

This led to the second approach to calculate a specific prediction equation for each of the exercises. The enumeration below lists the respective coefficients of the specific prediction equations for each exercise.


**Lat pulls:**
Regression coeffs: [9.87542505 × 10^−1^, 1.52700149 × 10^−2^, 2.78487038 × 10^−3^, −1.15638888 × 10^−4^]Isometric coefficient: 0.9992532650410967 (1RM Lat pulls/ISOM Lat pulls)



**Triceps dips:**
Regression coeffs: [9.73559954 × 10^−1^, 2.71642658 × 10^−2^, 9.30880743 × 10^−4^, −4.77453474 × 10^−5^]Isometric coefficient: 0.9992532650410967 (1RM triceps dips/ISOM triceps dips)



**Trunk extension**


Regression coeffs: [1.01565994e+00, −1.30808261 × 10^−2^, 6.01883266 × 10^−3^, −2.28623800 × 10^−4^]Isometric coefficient: 0.9992532650410967 (1RM trunk extension/ISOM trunk extens.)


**Trunk flexion:**
Regression coeffs: [9.94961852 × 10^−1^, 7.31240752 × 10^−3^, 2.93967456 × 10^−3^, −1.05109010 × 10^−4^]Isometric coefficient: 0.9992532650410967 (1RM trunk flexion/ISOM trunk flexion)



**Leg extension:**
Regression coeffs: [9.99714613 × 10^−1^, 1.84050532 × 10^−3^, 3.90281962 × 10^−3^, −1.28405344 × 10^−4^]Isometric coefficient: 0.9992532650410967 (1RM leg extension/ISOM leg extension)



**Leg flexion:**
Regression coeffs: [9.94637813 × 10^−1^, 7.15193846 × 10^−3^, 3.33621122 × 10^−3^, −1.09143358 × 10^−4^]Isometric coefficient: 0.9992532650410967 1RM leg flexion/ISOM leg flexion)


Table 5 provides the absolute values of the difference between actual 1RM and predicted 1RM for the six specific exercises.

Compared to the values of the leg press-based equation (Table 4), we observed non-significant absolute values for the difference between predicted and actual tests for most but not all exercises (Table 5). In parallel to the non-specific equation, r8 (i.e., 5–8RM) or r8 + isometric maximum were the most precise approaches to predict 1RM.

Of major importance, no significant gender differences were determined when comparing the predicted performance of specific prediction equations to the actual 1RM performances. More specifically, data for the most accurate r8 (5–8 RTF) or ISOM + r8 approach was (women) 3.45 ± 3.15% vs. (men) 4.13 ± 4.25% for the leg press, 4.11 ± 2.92% vs. 3.41 ± 2.44% for lat pulls, 5.83 ± 5.87% vs. 4.23 ± 5.11% for triceps dips, 5.21 ± 4.29% vs. 4.34 ± 0.78% vs. for trunk flexion, 5.10 ± 3.85% vs. 3.54 ± 3.83% for trunk extension, 5.28 ± 3.85% vs. 5.46 ± 4.18% for leg extensor and 6.83 ± 6.06 vs. 6.99 ± 4.47 for leg flexors.

## 4. Discussion

In the present study, the aim was to develop a 1RM strength prediction equation that accurately estimates 1RM from RTF for older people with osteopenia and osteoporosis generated on the proxomed compass 600 devices. The initial approach was to determine the performance accuracy of current prediction equations [12,13,14,15,16,17] to estimate 1RM from isometric maximum strength and a wide range of RTF for each exercise. Finally, accurate prediction equations for this vulnerable cohort of osteopenic older women and men were developed for each exercise.

In summary, the 1RM predicted by the cohort and machine-specific equation for the leg press and trunk flexion exercise were significantly more accurate compared with a present prediction equation dedicated to trained older people. Nevertheless, the available KLW [18] prediction equation provided acceptable prediction performance for most exercises; thus, we have to revise our hypothesis of the general superiority of specific prediction equations at least for this cohort and the present exercise. In parallel, although the cohort and machine-specific approach provided the most accurate prediction performance on average, differences between the specific prediction equation and the leg-press-based approach were non-significant (<0.7%). This indicates that the relevance of exercise exercise-specific prediction equation was low, at least in this cohort of older osteopenic adults and for the present exercise/device selection.

Results for the first part of the study found the KLW equation [18] adequately predicted 1RM using RTF for all but one exercise in the acceptable range of 4% to 6% of the absolute difference (Table 3). The KLW equation applied an “optimum cubic curve” that approximated all tests. The curve was fitted over four exercises (leg press, horizontal bench press, rowing, leg adduction). Unlike other available equations [12,15,21,27], the KLW prediction equation calculated 1RM over a wider RTF range (3–20 reps). In addition, focusing equations on an older osteopenic cohort with a history of RT [28] and applying exercises on different resistance devices may explain the similarity of prediction performance. In summary, the newly developed equations predicted 1RM from RTF slightly more accurately, with significant differences only for trunk flexion and leg press exercise. Thus, prediction equations calculated specifically for each exercise did not really outperform the leg press-based approach.

Therefore, the question is whether it is worth the effort to calculate a prediction formula for each exercise performed during an RT program. This question must be answered by each trainer or athlete for themselves. Nevertheless, the decision might strongly depend on the relevance of precise training specifications for achieving a unique training goal, which is certainly higher in the athletic area than for fitness and health-orientated participants. Nonetheless, in older people with osteopenia and osteoporosis, the relevance of adequate RT intensity is crucial to prevent fractures from mechanical overload. Therefore, we preferred to apply the most accurate option to determine 1RM.

In the present study, the 5–8 RTF range was the most accurate range to predict 1RM. For higher repetition ranges (i.e., r10, r15, Table 4 and Table 5), we observed rather inconsistent results varying from accurate prediction performance (e.g., lat pulls, triceps dips) to non-reliable data (trunk flexion, leg extension, leg flexion). Correspondingly, RTF tests should be restricted to periods of intensive RT (i.e., periods with low number of repetitions and repetition to fatigue), at least for the latter exercises. This inconsistent finding is in contrast to the KLW approach [18] that reported comparable and, in summary, more accurate prediction performance (1.5% to 3.2%) for all RTF ranges (3–5, 6–10, 11–15 and 16–20 reps) independent of the specific exercise. As a novelty in the present approach, we included the isometric maximum which improved prediction performance in three of seven exercises (Table 4 and Table 5), but it does prevent the application of these equations for exercises without the possibility of isometric testing. Thus, with reference to the present study, the 5–8 RTF approach might be the most accurate solution to predict 1RM. One may argue that lower repetition ranges (i.e., 2–4 RTF) might provide even better prediction performance. However, 2–4 reps to failure is a range rarely specified by exercise protocols for older adults, be it with or without osteopenia/osteoporosis.

As mentioned previously, when including the isometric maximum strength test, the predictive power of specific prediction equations ranges between 3.7% and 6.9% (absolute value). We conclude that this predictive performance will be sufficiently accurate to reliably specify exercise intensity using this equation. This estimation is based on the coefficients of variation (CV) of the 1RM that average between 3.3% to 5.4% and on the reproducibility of the 5–8 RTF range in this cohort (in average 0.9 ± 1.0 reps). Considering that one repetition performed more or less contributed to approximately a 2% change in 1RM, this might indicate the relevance of normal deviations in individual performance for the accuracy of prediction equations.

The present project possesses some strong points. As one positive aspect of the study, we focused on strict standardization of the 1RM and RFT tests by specifying the identical speed of movement, starting angle, and lifting pattern which were strictly supervised by research assistants. Further, we included a cohort of people with considerable training experience who were able to reliably conduct 1RM and RTF tests on RT machines. As checked and queried by the research assistants, compliance with the 1RM and RTF was very high in this cohort. Reproducibility of the 1RM tests (10 women and men each) averaged in the range of CVs listed for trained young and middle-aged cohorts [29]. Considering further that due to our blinding approach, participants did not know what load they were moving during the 1RM test, we feel our data on 1RM CVs are particularly meaningful. While no test-retest reliability data are available for the RTF tests, the differences of ±1 repetition among RFT tests in the 5–8 rep-ranges suggest strong reproducibility. We focus on older people with osteopenia and osteoporosis, i.e., people with increased fracture risk. As already mentioned, so far, no study has reported study fractures during 1RM testing; however, only very few studies with osteopenic people applied regular 1RM tests to specify exercise intensity (e.g., [30]). Undoubtedly, fracture risk increases when applying maximum loads—at least for some exercises, including in particular trunk extension or flexion. Assessing 1RM by sub-maximum tests decreases the low but crucial risk of fractures and might thus increase the acceptance of staff and participants for regular strength testing procedures. Apart from their “bone status”, the training status of the participants was very homogeneous. All participants had taken part in previous exercise studies (e.g., [23,24,25]) and have continued their fracture prevention exercise training in our university health sports club for a minimum of 3 years so far. The exercise protocol, which includes 60 min of weight-bearing and resistance exercise twice a week, was roughly identical for all participants included in this trial.

However, some limitations of the study also have to be considered. Firstly, the statistical power of the study was affected by the limited sample size (n = 40) which was insufficient to conduct more dedicated subgroup analyses apart from gender differences (e.g., age groups, training experience, health status). Secondly, we do not apply a randomized sequence of exercise testing but use a standardized order. Should our approach of not continuing 1RM (or RTF) before participants stated full recovery have failed and hence cumulated fatigue confounded our results, this standardized sequence would have to be considered as a limitation. Further, although we consistently asked participants about “full recovery” prior to the next attempt, exercise, or testing session we cannot be sure whether we fully excluded accumulated fatigue with a negative impact on our findings.

An even more important issue is whether the prediction equation derived by the proxomed compass 600 devices can be properly transferred to other resistance exercise training machines or even to free-weight exercises. The specificity of the proxomed compass 600 series predominately refers to exact positioning, safety aspects, software application and isometric testing mode, while movement and lifting procedures apply conventional techniques (Figure 1). Considering the device comprehensive relationship between 1RM and RTF(s), we feel that our prediction equation is transferable to other devices using conventional movement and lifting techniques. This rating is confirmed by the finding that the KLW prediction equation calculated for osteopenic postmenopausal women [18] based on Techno Gym equipment (Gambettola, Italy) but accurately predicted 1RM for the present approach. However, due to enhanced neuromuscular abilities, we think the accuracy of our prediction equation is reduced when transferred to free-weight exercises. Nevertheless, we feel that participant motivation and test compliance are the most important confounding factors to accurately predicting 1RM from RTF in non-athletic cohorts.

We included men and women in our evaluation in order to extend the generalization of our prediction equation. Comparing the sexes, we observed minor and undirected differences; thus, we concluded that, at least in our cohort of trained older people, sex has a negligible effect on prediction performance. Considering further, that there is some evidence [8,29], that the reproducibility and validity of 1RM testing were not significantly affected by RT experience, transferring application of these RTF equations to non-athletic cohorts seems reasonable. Thus, we suggest the expansion of the current 1RM and RTF testing protocol to other cohorts of 60-year-olds and older.

## Figures and Tables

**Figure 1 sports-12-00233-f001:**
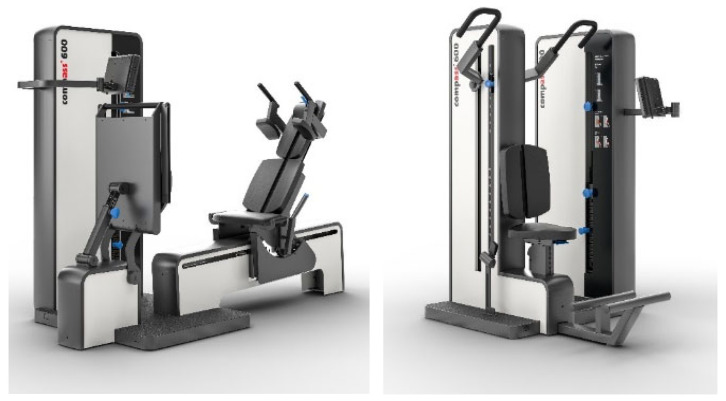
Resistance training devices used for the present study. Leg press (**top line on the right**), lat pulls/triceps dips (**top line on the left**), trunk extension/-flexion (**bottom line left**) and leg extension/-flexion (**bottom line right**).

**Figure 2 sports-12-00233-f002:**
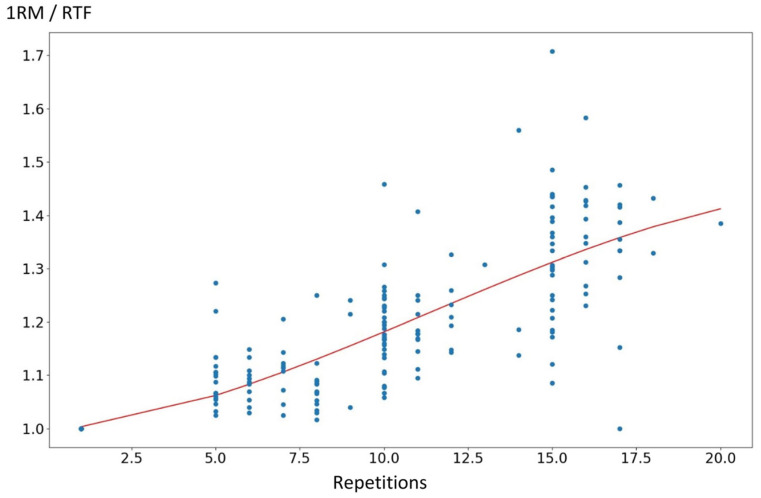
Polynominal regression equation for the leg press exercise in trained older adults with osteoepenia. 1RM: repetition maximum; RTF: repetition.

**Table 1 sports-12-00233-t001:** Prediction equations applied in the present analysis; wt: load, reps: number of repetitions.

Brzycki [12]	1RM = rep weight/(1.0278 − (0.0278 × reps))
Epley [13]	1RM = (1 + 0.333 × reps) × rep wt
Lander [14]	1RM = 100 × rep wt/101.3 − 2.26123 × reps
Mayhew et al. [15]	1RM = 100 × rep wt/52.2 + 41.9 × exp [−0.055 × reps]
O’Connor et al. [16]	1RM = rep wt (1 + 0.025 × reps)
Wathan [17]	1RM = 100 × rep wt/(48.8 + 53.8 × exp [−0.075 × reps]
KLW [18]	1RM = (rep wt [0.988 − 0.0000584 × reps^3^ + 0.0019 × reps^2^ + 0.0104 × reps]

**Table 2 sports-12-00233-t002:** Participant characteristics at baseline.

Variable	Men (n = 17) Mean ± SD	Women (n = 23) Mean ± SD	Total Cohort Mean ± SD
Age [years]	73.4 ± 9.0	71.4 ± 7.8	72.8 ± 7.4
Body mass [kg]	80.1 ± 8.1	63.8 ± 10.4	70.7 ± 12.9
Body height [cm]	176.0 ± 5.8	165.0 ± 8.2	169.7 ± 10.2
Total bodyfat [%]	23.7 ± 3.3	30.1 ± 3.3	27.4 ± 3.9
Osteopenia/Osteoporosis [n]	14/2	13/7	13/7
BMD T-Score lumbar spine (SD)	2.1 ± 0.5	2.3 ± 0.8	2.2 ± 0.6
BMD T-Score total hip (SD)	2.0 ± 0.9	2.1 ± 1.0	2.1 ± 1.0
Cardiometabolic diseases [n]	6	8	14
Resistance training history [years]	9.6 ± 2.9	13.1 ± 5.1	11.6 ± 6.4

BMD: Bone Mineral Density.

**Table 3 sports-12-00233-t003:** MV and SD of the predicted 1RM (upper line) and the absolute values of the difference between actual 1RM and predicted 1RM of seven prediction equations.

Exercise	1RM Test	Brzycki Predicted 1RM Rel. Diff. (%)	Epley Predicted 1RM Rel. Diff. (%)	Lander Predicted 1RM Rel. Diff. (%)	Mayhew Predicted 1RM Rel. Diff. (%)	O’Connor Predicted 1RM Rel. Diff. (%)	Wathan Predicted 1RM Rel. Diff. (%)	KLW Predicted 1RM Rel. Diff. (%)
Leg press [kg]	129.5 ± 41.1	140.6 ± 46.8 *** 9.1 ± 6.5	144.8 ± 48.0 *** 11.9 ± 6.6	141.9 ± 47.2 *** 9.9 ± 6.7	146.3 ± 48.2 *** 12.9 ± 6.1	138.6 ± 45.7 *** 7.6 ± 5.1	145.2 ± 48.3 *** 12.3 ± 6.8	133.7 ± 44.1 ** 5.4 ± 3.3
Lat pulls [kg]	80.9 ± 30.3	82.1 ± 30.6 5.1 ± 4.9	84.9 ± 31.6 *** 6.7 ± 5.3	82.9 ± 30.9 5.4 ± 5.1	86.2 ± 32.0 *** 7.8 ± 5.3	81.6 ± 30.3 4.7 ± 4.1	85.0 ± 31.7 *** 6.9 ± 5.5	78.5 ± 29.1 * 4.4 ± 4.1
Triceps dips [kg]	116.9 ± 42.8	118.8 ± 45.8 5.3 ± 3.6	122.8 ± 47.5 *** 6.7 ± 4.3	120.0 ± 46.3 * 5.5 ± 3.9	124.7 ± 48.4 *** 7.8 ± 4.5	118.0 ± 45.7 5.3 ± 3.3	122.9 ± 47.5 *** 6.8 ± 4.4	113.5 ± 43.9 * 5.2 ± 4.3
Trunk flexion [kg]	48.6 ± 17.7	47.6 ± 19.6 9.0 ± 8.9	49.1 ± 20.3 9.6 ± 8.4	48.1 ± 19.8 8.9 ± 8.6	50.4 ± 20.6 9.2 ± 7.9	47.6 ± 19.5 8.2 ± 8.3	48.9 ± 20.4 10.3 ± 8.8	47.1 ± 17.7 8.1 ± 8.8
Trunk extension [kg]	71.2 ± 19.2	74.9 ± 21.0 *** 6.2 ± 5.1	77.1 ± 21.5 *** 8.1 ± 5.9	75.6 ± 21.2 *** 6.6 ± 5.5	78.0 ± 21.6 *** 9.3 ± 5.1	73.9 ± 20.5 *** 4.7 ± 3.9	77.3 ± 21.6 *** 8.3 ± 6.4	71.3 ± 19.8 4.1 ± 2.8
Leg extension [kg]	81.8 ± 18.2	85.3 ± 21.7 * 6.8 ± 5.9	87.9 ± 22.3 *** 8.9 ± 5.9	86.1 ± 21.9 ** 7.5 ± 5.9	89.0 ± 22.5 *** 9.9 ± 5.5	84.3 ± 21.3 * 6.0 ± 5.0	88.2 ± 22.4 *** 9.2 ± 6.2	81.2 ± 20.6 5.6 ± 4.7
Leg flexion [kg]	62.8 ± 22.0	65.9 ± 25.1 ** 8.3 ± 6.0	67.9 ± 25.8 *** 9.9 ± 6.2	66.6 ± 25.4 ** 8.8 ± 6.0	68.5 ± 25.9 *** 10.4 ± 6.0	64.9 ± 24.6 * 7.0 ± 5.3	68.1 ± 25.9 *** 10.3 ± 6.5	62.6 ± 23.7 6.2 ± 5.5

*** *p* ≤ 0.001; ** *p* ≤ 0.01; * *p* ≤ 0.001 (difference predicted vs. actual 1RM).

**Table 4 sports-12-00233-t004:** MV and SD of the absolute values of the difference between actual 1RM and predicted 1RM. Calculation based on the approximation equation computed for the leg press exercise.

	Leg Press	Lat Pulls	Triceps Dips	Trunk Extension	Trunk Flexion	Leg Extension	Leg Flexion
RTF	Rel. Diff. ± SD (%)	Rel. Diff. ± SD (%)	Rel. Diff. ± SD (%)	Rel. Diff. ± SD (%)	Rel. Diff. ± SD (%)	Rel. Diff. ± SD (%)	Rel. Diff. ± SD (%)
r15	6.52 ± 5.23	10.29 ± 5.87	8.22 ± 4.65	10.72 ± 6.33	11.5 ± 11.9	13.3 ± 10.7	13.9 ± 10.3
r10	4.39 ± 3.77	8.66 ± 5.25	8.49 ± 4.87	6.38 ± 4.60	8.51 ± 7.85	9.55 ± 7.83	9.69 ± 8.43
r8	4.45 ± 3.07	5.95 ± 4.43	6.30 ± 5.14	4.33 ± 2.49	5.28 ± 4.39	5.99 ± 4.77	6.38 ± 5.71
ISOM	6.97 ± 6.28	6.90 ± 6.67	12.7 ± 12,8	13.5 ± 10.8	19.6 ± 14.7	19.1 ± 12.3	12.6 ± 11.4
ISOM + r8	3.72 ± 3.65	3.70 ± 2.29	5.95 ± 6.30	6.41 ± 5.26	9.52 ± 7.95	8.48 ± 6.20	7.44 ± 6.39

**Table 5 sports-12-00233-t005:** MV and SD of the absolute values of the difference between actual 1RM and predicted 1RM based on the approximation equation specifically computed for each exercise ^1^.

	Lat Pulls	Triceps Dips	Trunk Extension	Trunk Flexion	Leg Extension	Leg Flexion
RTF	Rel. Diff. ± SD (%)	Rel. Diff. ± SD (%)	Rel. Diff. ± SD (%)	Rel. Diff. ± SD (%)	Rel. Diff. ± SD (%)	Rel. Diff. ± SD (%)
r15	6.11 ± 5.15	5.65 ± 4.64	8.22 ± 5.82	12.59 ± 9.89	13.00 ± 8.52	11.29 ± 8.06
r10	5.53 ± 4.17	5.77 ± 4.43	6.06 ± 4.39	9.94 ± 5.66	8.59 ± 6.36	8.76 ± 6.44
r8	5.00 ± 4.18	5.73 ± 3,33	4.59 ± 2.93	4.82 ± 3.25	5.41 ± 5.18	6.90 ± 5.45
ISOM	5.75 ± 5.32	7.91 ± 9.92	9.86 ± 9.18	15.5 ± 14.0	8.38 ± 6.28	13.8 ± 10.8
ISOM + r8	3.82 ± 2.76	5.23 ± 5.66	5.75 ± 4.41	8.38 ± 7.87	5.34 ± 3.97	8.67 ± 5.92

^1^ Data for the leg press exercise is listed in Table 4.

## Data Availability

The raw data supporting the conclusions of this article will be made available by the authors on request due to privacy issues.

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
