# Peer review of "Developing Accurate Repetition Prediction Equations for Trained Older Adults with Osteopenia"

_sports, 2024, doi:10.3390/sports12090233_

Round 1

Reviewer 1 Report

Comments and Suggestions for Authors

Developing Accurate Repetition Prediction Equations for Trained Older Adults with Osteopenia

Abstract

You say “selected machines” yet specifically pre-selected a machine that you want to validate. Be specific with the machine choice.

What is the “one” previously acceptable equation? Be specific.

Need to discuss the acceptable coefficient of variation for RTF to 1RM in the introduction.

Introduction

Need to discuss and give information on what you consider “older adults”. Literature on this in exercise science is very robust.

Line 42: Add “to strength” after bone adaptation.

Line 44: “Older people” this manuscript is on older people with osteoporosis.

Line 46-50: Break this sentence up and reword for clarity.

Your purpose statement is to try to predict these equations in very specific machines chosen by you. Therefore, your purpose has to be to that specific machine. “-This project is part of an ongoing investigation aimed at gathering normative values 62 for the proxomed compass 600 (Alzenau, Germany) resistance exercise device.”

Methods

Be specific on how many years of resistance training experience.

What was the warm up protocol?

What valid protocol for rest intervals between tests did you use?

How can a movement be concentric on both sides of the isometric? Provide the reference for your dynamic ECC/iso/con time.

Results

Please provide the participant mean +/- SD t-score in table 2.

Discussion

First sentence: on a specific machine.

Limitations exist due to the specificity to your machine choice, and whether it is transferable to free weights and other machines.

Do not include study “strengths” in the limitations section. It is ok to be honest and upfront about the several study limitations to enhance the field and future research.

Comments on the Quality of English Language

Seems fine. 

Reviewer 2 Report

Comments and Suggestions for Authors

Brief Summary – Aim of Paper and Strengths This manuscript sought to define and improved equations for estimation of a 1-repetition maximum from various repetition-to-failure rep ranges. This paper also sought to delineate if calculating predictive equations for individual exercises is more accurate than solely using leg press performance for these predictive equations. Specifically, these equations are aimed to optimize training in older people who have resistance training experience and osteopenia or osteoporosis. Though the authors make a case for these equations being generalizable to untrained populations. In either case, these populations hugely benefit from resistance exercise; understanding what relative-loading scheme is appropriate for these individuals could provide further benefit by improving training efficiency and reducing relative risk. Through this manuscript the authors use their data set to create an equation that more accurately predicts 1-RM based on several rep-ranges and several exercises. They have also done their due diligence in comparing existing prediction equations to their 1-RM versus RTF data set. In addition, a strength of this manuscript is in the control of exercise parameters, thus making the testing more reproducible and generalizable to non-trained populations. Although, it is important to note that these equations will likely be less accurate in populations with different ages, training ages, exercise back grounds (e.g., endurance training or powerlifting), progression into their training cycles, and experience utilizing these specific devices. This reviewer suggests that the addition of perceived difficulty ratings may have benefited the generalization of this equation. Often practitioners will use prediction equations as a starting point to select relative load, and them must use in-training performance and rating of perceived difficulty to apply progressive over-load. Knowing the exact starting load, therefore does not benefit the selection of weights after a few weeks of progressive overload training. This reviewer does acknowledge that these subjective ratings can often vary wildly depending on the adeptness of participants to training and understanding of physical ability. Hopefully, in the future, these authors can address predictive equations used in axially loaded RT exercises and more complex movements that do not have a fixed range of motion. Of course, these are more complex to experimentally control in this population but are of critical importance to older peoples with osteopenia/osteoporosis. Abstract Line 18: Be careful of using acronyms only once (e.g., MIS -Isometric Maximum Strength) Introduction Line 39: I agree proper specification of exercise intensity is critical in designing training programs, but perhaps mention how this is important in the context of exercise volume and specificity (i.e., fitness characteristics like strength, hypertrophy, endurance, etc.) Line 41: “fracture risks”: Later in this section you mention evidence that 1RM testing is not contra-indicated for people with osteopenia/osteoporosis. It stands to reason then, that acute application of training load is not as important as chronic application. Therefore, fatigue, tissue damage, and recovery are more important to manage than load in the context of improving the outcome for people with bone strength deficiencies. To this point I suggest a greater emphasis on how practitioners can use predictive equations to better manage training volume (i.e., Sets x Reps X Weight) on a weekly to monthly basis. Line 49: “interruption of the continuity of the training 49 process aggravate the frequent application of 1RM tests particularly in non-athletic RT.” The wording of this sentence makes it difficult to understand what you mean to convey. If I understand, I think you mean that by eliminating the need to conduct 1-RM testing sessions, then more time could be spent on training itself. This is an interesting proposition, with obvious benefits to athletic and non-athletic populations; often performance testing is conducted too frequently. Line 52: “option to 1RM testing” I suggest adding the word “alternative” in front of “option”. Line 53-56: In this section you mention equations existing for older adults. Could you confirm if there are any current equations for trained older adults? Likewise, could you add a rational for why you believe people with osteopenia/osteoporosis would potentially differ in their predictive equations? In trained older adults, for example, would you predict 1RM to be significantly different between those with and without osteopenia? Methods Lines 75-79 “Eligibility criteria for inclusion were (1) age 60 years and older, (2) osteopenia or osteoporosis at the lumbar spine or femoral neck as determined by Dual-Energy X-ray Absorptiometry [25], (3) continuous history of resistance exercise for at least five years, and (4) experience in 1RM test procedures and repetition to fatigue (RTF) tests.” This description is good, but this reviewer feels a couple of items need clarification. First, what was the timeframe of diagnosis? Was this a report from a physician and if so, how long ago were the participants diagnosed? Did the research team confirm presence of osteopenia/osteoporosis? If you were able to confirm by DEXA, please provide methods for this. Second, for training history, was there any additional information collected about the characteristics of their typical training? This would make a large difference in the neuromuscular adaptations incurred from their training and thus the validity of your predictive equation. Third, while participants were, “familiar” with 1RM and RTF testing, were they familiarized with the exact parameters and procedure for this protocol? Last, did participant begin training before or in response to their osteopenia/ osteoporosis diagnosis? Section 2.2 Testing Procedures Please describe if these tests were conducted and supervised by qualified personnel. Do they hold certification from a governing body (e.g., National Strength and Conditioning Association)? Additionally, please provide the method for isometric testing. Include what the units of measure were and the angle that the tests were conducted at. Lines 98- 101: Were exercises selected in a randomized order, with exception for similar muscle groups being spaced? Further, what is your rational for not randomizing 1RM and RTF tests. The current program would suggest exercises conducted in session 4 may experience fatigue from the previous three weeks of max testing. Line 105-107: What was the method to determine how much the weight should be increased upon successful maximum attempts, and how did you determine that 4-5 attempts was acceptable in this population? Line 108-109: This section should be reworded. As it reads now, I understand it to mean you conducted RTF tests based on a percentage of 1RM. Which you would need to cite where you found this information. Alternatively, maybe this means you just noted the percentage that successful RTF tests were in comparison to their paired 1RM tests? Line 113-117: Please indicate whether these tests were considered inter- or intra- rater reliability tests. Line 118: Does the manufacturer detail indicate the use of anatomical landmarks to determine the proper scaling of the machine? Line 124-125: Please re-write this description. This reads as if you conducted 1RM tests with loads equal to 5-8RM, 9-12RM and 13-16RM, then did RTF in the same rep ranges. If this is the case, I did not understand such from the methods above. Line 130: What was the rationale for choosing the 5-8RTF range? Table 1: the equation from Bryzcki should read as 1RM = rep weight/ (1.0278 – (0.0278 reps)). Line 137: Do you mean this to be the Wilcoxon Sign-ranked test or a paired t-test only. If the Wilcoxon test is indicated, please describe why you used non-parametric statistics and where and how the distribution is non-normal. Line 138: Please indicate if this is intended to match the non-parametric statics listed above. Results Line 151-152: “It is worth noting that no injuries 151 or adverse effects were observed or reported after the 300 1RM and 800 RTF tests.” It is great that there were no injuries, and this reviewer appreciates your inclusion of that statement, but the totaling the number of tests is not necessary for the paper. Table 2: Would it be possible to add a column of both males and females averaged, in the results you perform analyses on these groups combined, therefore it would aid in interpretation. Table 3: This may just be an uploading/downloading issue, but in the exercise column, “[kg]” should be moved up to sit next to the exercise name. Line 166 – 182: This section should be reorganized. IT is difficult to read the equation and understand what when into it. I suggest the addition of an equation figure with reorganization of what’s to be included. Likewise for Lines: 195-210 Line 195: It is not clear what the Isometric co-efficient is or how this was calculated. Discussion Line 233-238: This section seems to be contradictory. It reads that in this population exercise specific prediction equations are superior, but the difference between exercise specific and legpress based prediction equations are negligible. This would suggest that the adoption of exercise specific prediction equations is arbitrary. Line 248: Do you think this is a coincidence or does the diagnosis of osteopenia/osteoporosis in the spine and femoral head play a role in leg-press and trunk flexion exercises not being accurately predicted? Line 262: What is meant by “restricted to periods of intensive RT”? Line 269: Why not look at rep ranges in the 2-4 RTF, from the evidence laid out, predictions get more accurate the closer you get to 1RM. It would also be more specific to the energy systems used during 1RM testing. Of course, the relevance of 1RM testing is determined by the specific goals of each person. Line 289-295: The reproducibility of rep max testing is likely the most crucial aspect of this study. In this section, it is not totally clear if you mean the 1RM CVs from this data set match cohorts from another study (this is what I think you mean) or if the 1RM CVs you used were data from another data set. Also, could you further explain the reproducibility data from RTF tests, where they differ by +/- 1. Did participants perform an additional set if they only completed 4 reps? From your methods, I understand that by performing 9 reps on the 5-8RTF session, that data instead went to the 9- 12RTF rep range and the 5-8RTF was trialed again. In accordance with my previous comment here, do you think giving participants the ability to redo the 5-8RTF aided in the increased accuracy? Line 299-300: “Considering further the present evidence that reproducibility and validity of 1RM testing are not dependent on RT experience [8, 29], transferring application of these RTF equations to non-athletic cohorts seems appropriate.” This reviewer suggests being careful of the language used here. To say that the validity and reproducibility of 1RM testing are not dependent on RT experience is likely too broad. This stamen may hold true here, but could quickly become false (e.g., different exercises, different age groups, etc.).

Author Response

Dear Reviewer,

Thank you very much for your careful review of the article. We hope we have adequately answered all the questions and addressed all the suggestions and concerns. The changes and amendments are highlighted in the text and are thus easily be identified. Please find below the answers to the specific comments and the changes arising from the recommendations.

Brief Summary – Aim of Paper and Strengths

This manuscript sought to define and improved equations for estimation of a 1-repetition

maximum from various repetition-to-failure rep ranges. This paper also sought to delineate if

calculating predictive equations for individual exercises is more accurate than solely using leg

press performance for these predictive equations. Specifically, these equations are aimed to

optimize training in older people who have resistance training experience and osteopenia or

osteoporosis. Though the authors make a case for these equations being generalizable to

untrained populations. In either case, these populations hugely benefit from resistance exercise;

understanding what relative-loading scheme is appropriate for these individuals could provide

further benefit by improving training efficiency and reducing relative risk.

Through this manuscript the authors use their data set to create an equation that more

accurately predicts 1-RM based on several rep-ranges and several exercises. They have also done

their due diligence in comparing existing prediction equations to their 1-RM versus RTF data set.

In addition, a strength of this manuscript is in the control of exercise parameters, thus making the

testing more reproducible and generalizable to non-trained populations. Although, it is important

to note that these equations will likely be less accurate in populations with different ages,

training ages, exercise back grounds (e.g., endurance training or powerlifting), progression into

their training cycles, and experience utilizing these specific devices.

This reviewer suggests that the addition of perceived difficulty ratings may have

benefited the generalization of this equation. Often practitioners will use prediction equations as

a starting point to select relative load, and them must use in-training performance and rating of

perceived difficulty to apply progressive over-load. Knowing the exact starting load, therefore

does not benefit the selection of weights after a few weeks of progressive overload training. This

reviewer does acknowledge that these subjective ratings can often vary wildly depending on the

adeptness of participants to training and understanding of physical ability.

Hopefully, in the future, these authors can address predictive equations used in axially

loaded RT exercises and more complex movements that do not have a fixed range of motion. Of

course, these are more complex to experimentally control in this population but are of critical

importance to older peoples with osteopenia/osteoporosis.

General note.

The present article was initially designed as a “short research report”, that is why we have kept it very short at many places. 

Abstract

Line 18: Be careful of using acronyms only once (e.g., MIS -Isometric Maximum Strength)

We agree, we now have consistently written out “isometric maximum strength” and delete “MIS”.

Introduction

Line 39: I agree proper specification of exercise intensity is critical in designing training

programs, but perhaps mention how this is important in the context of exercise volume and

specificity (i.e., fitness characteristics like strength, hypertrophy, endurance, etc.).

Thank you for this recommendation. We now have added information to this issue.

Line 41: “fracture risks”: Later in this section you mention evidence that 1RM testing is not

contra-indicated for people with osteopenia/osteoporosis. It stands to reason then, that acute

application of training load is not as important as chronic application. Therefore, fatigue, tissue

damage, and recovery are more important to manage than load in the context of improving the

outcome for people with bone strength deficiencies.

Here, we do not completely agree. Both aspects, adequate acute load, be it generated by strain magnitude or -rate, -frequency or -duration, and chronic aspects related to adequate recovery and adaptation are similarly important for successful and specific bone adaptation. Inadequate acute strain might generate either no adaptative response when below bones adaptive threshold or damage bone when above the area of micro-fractures (≥3000 µΣ) or bones fracture threshold (≥25.000 µΣ). Indeed, it would be interesting to discuss the issue of bones specific response to different types of exercise, exercise parameters and training principles. However, in the present article we simply focus on developing a 1RM prediction equation that determines appropriate prescribed load for different repetition ranges in older adults with osteopenia/osteoporosis.

To this point I suggest a greater emphasis on how practitioners can use predictive equations to

better manage training volume (i.e., Sets x Reps X Weight) on a weekly to monthly basis.

We now have added this aspect to emphasize the relevance of predictive equations to better manage training.  However, we think this did not refer exclusively to training volume of the resistance exercise but even more to aspects related to individualization, specificity, progression and periodization.

Line 49: “interruption of the continuity of the training 49 process aggravate the frequent

application of 1RM tests particularly in non-athletic RT.”

The wording of this sentence makes it difficult to understand what you mean to convey. If I

understand, I think you mean that by eliminating the need to conduct 1-RM testing sessions, then

more time could be spent on training itself. This is an interesting proposition, with obvious

benefits to athletic and non-athletic populations; often performance testing is conducted too

frequently.

The wording of this sentence makes it difficult to understand what you mean to convey. If I

understand, I think you mean that by eliminating the need to conduct 1-RM testing sessions, then

more time could be spent on training itself. This is an interesting proposition, with obvious

benefits to athletic and non-athletic populations; often performance testing is conducted too

frequently.

Thank you for this recommendation. We now have revised the sentence for more clarity.

Due to the possibility to conduct RTF-Tests in different repetition ranges (however most accurately in the range of 5-10 reps), a very dense testing approach can be introduced. The only difference to “normal” RT sessions will be that repetitions have to be performed to failure.  

Line 52: “option to 1RM testing”

I suggest adding the word “alternative” in front of “option”.

Thank you for this recommendation: We now have replaced “option” by “alternative”.

Line 53-56: In this section you mention equations existing for older adults. Could you confirm if

there are any current equations for trained older adults? Likewise, could you add a rational for

why you believe people with osteopenia/osteoporosis would potentially differ in their predictive

equations? In trained older adults, for example, would you predict 1RM to be significantly

different between those with and without osteopenia?

There is a prediction equation for postmenopausal women with osteopenia in average 57±3 years old (Kemmler et al., 2006). However, apart from differences in age this available 1RM prediction equation was developed using other devices and partially other exercises. As stated, in summary, the prognostic performance of this approach generated usable values for the present cohort, at least for most devices or exercises. Nevertheless, we aimed to develop a more accurate approximation equation that fits all exercises and devices.

Methods

Lines 75-79

“Eligibility criteria for inclusion were (1) age 60 years and older, (2) osteopenia or osteoporosis

at the lumbar spine or femoral neck as determined by Dual-Energy X-ray Absorptiometry [25],

(3) continuous history of resistance exercise for at least five years, and (4) experience in 1RM

test procedures and repetition to fatigue (RTF) tests.”

This description is good, but this reviewer feels a couple of items need clarification. First, what

was the timeframe of diagnosis? Was this a report from a physician and if so, how long ago were

the participants diagnosed? Did the research team confirm presence of osteopenia/osteoporosis?

If you were able to confirm by DEXA, please provide methods for this.

We now have added BMD values in Table 2. We further give more details on the procedures to determine the osteopenia osteoporosis statement. As mentioned all of the participants were participants of foregoing studies (Hettchen et al., 2021; Kemmler et al., 2015; Kemmler et al., 2020) on exercise and bone strength. Thus, we have extensive data on this cohort.

Second, for training history, was there any additional information collected about the

characteristics of their typical training? This would make a large difference in the neuromuscular

adaptations incurred from their training and thus the validity of your predictive equation.

In addition to resistance exercises we applied also weight bearing exercises with diverging strain rate, dependent on the status of the individuum. Apart from the joint “anti-fracture exercise program” in our health sport club, few people performed additional exercise, predominately walking, cycling or swimming.  We do not think that these types of exercises produce large differences in neuromuscular adaptations related to resistance exercise.

Third, while participants were, “familiar” with 1RM and RTF testing, were they familiarized

with the exact parameters and procedure for this protocol?

Our usual training protocol is based on frequent RTF tests and 1 RM-tests twice a year, however, has not yet performed on the proxomed compass 600 resistance training devices. However, apart from slight differences in individual positioning and the isometric testing procedure, the general approach of 1RM and RTF tests did not differ between the usual tests applied with the Techno Gym equipment and the proxomed compass 600 devices.

Last, did participant begin training before or in response to their osteopenia/ osteoporosis

diagnosis?

Due to the eligibility criteria for the fore going studies (Hettchen et al., 2021; Kemmler et al., 2015; Kemmler et al., 2020) only people with a diagnosis of osteopenia/ osteoporosis were included. Thus, participants started training in response to their osteopenia/ osteoporosis diagnosis.

Section 2.2 Testing Procedures

Please describe if these tests were conducted and supervised by qualified personnel. Do they

hold certification from a governing body (e.g., National Strength and Conditioning Association)?

We now have added information on qualification of the research assistants that supervised and guided the 1RM and RTF-tests.

Additionally, please provide the method for isometric testing. Include what the units of measure

were and the angle that the tests were conducted at.

 The units are indicated by the device in “Newton”. A section with detailed information on participant’s positioning, device settings and joint angles of the isometric and dynamic testing procedures were added.

Lines 98- 101: Were exercises selected in a randomized order, with exception for similar muscle

groups being spaced? Further, what is your rational for not randomizing 1RM and RTF tests.

We do not use a randomization order for the sequence of the exercises or tests. Considering that exercises for related muscle groups or agonist/antagonist in immediate succession were not intended there is rather limited potential for a proper randomization procedure. With respect to RTF tests we do not see a rationale for randomization – what would be the benefit of this approach?

The current program would suggest exercises conducted in session 4 may experience fatigue from the previous three weeks of max testing.

Thank you for this hint. However, we do not agree that with 7 days of recovery between the test sessions a relevant accumulation of fatigue will confound the test results – at least in this resistance trained cohort.

Line 105-107: What was the method to determine how much the weight should be increased

upon successful maximum attempts, and how did you determine that 4-5 attempts was acceptable

in this population?

As given our test assistants/trainers and the participants are (very) familiar with 1RM and RTF tests. Weight increases after successful attempts were conducted in close cooperation between participants and research assistants. This is our standard procedure during 1 RM tests.

Line 108-109: This section should be reworded. As it reads now, I understand it to mean you

conducted RTF tests based on a percentage of 1RM. Which you would need to cite where you

found this information. Alternatively, maybe this means you just noted the percentage that

successful RTF tests were in comparison to their paired 1RM tests?

Thank you for the recommendation. We now have clarified the sentence and explained the approach in more detail including a reference.

Line 113-117: Please indicate whether these tests were considered inter- or intra- rater reliability

tests.

Thank you for the recommendation. We now have added the type of the test.

Line 118: Does the manufacturer detail indicate the use of anatomical landmarks to determine

the proper scaling of the machine?

Yes. Further, we now have prescribed participant positioning, device settings and range of motion in much more detail. Thus, we think the reader can adequately understand the testing procedures.

Line 124-125: Please re-write this description. This reads as if you conducted 1RM tests with

loads equal to 5-8RM, 9-12RM and 13-16RM, then did RTF in the same rep ranges. If this is the

case, I did not understand such from the methods above.

Sorry for this mistake, the sentence was indeed not comprehensible and was revised for clarity.

Line 130: What was the rationale for choosing the 5-8RTF range?

We aimed to generate a uniform distribution of the number of repetitions as the basis for the formula calculation, whereby the range of repetitions within the range should not be too high. In addition, the range of 5-8 repetitions is frequently addressed and can be easily implemented by the participants.

Table 1: the equation from Bryzcki should read as 1RM = rep weight/ (1.0278 – (0.0278 reps)).

Thank you, we now have revised the equation

Line 137: Do you mean this to be the Wilcoxon Sign-ranked test or a paired t-test only. If the

Wilcoxon test is indicated, please describe why you used non-parametric statistics and where and

how the distribution is non-normal.

Sorry for this spelling mistake: We used the Wilcoxon paired sample test or the paired t-test where applicable. Normal distribution was determined using Q-Q plots and the Shapiro Wilk Tests. The majority of the data was normally distributed.

Line 138: Please indicate if this is intended to match the non-parametric statics listed above.

We now have added the hint that this parameter was normally distributed. Thank you for the recommendation.

Results

Line 151-152: “It is worth noting that no injuries or adverse effects were observed or

reported after the 300 1RM and 800 RTF tests.”

It is great that there were no injuries, and this reviewer appreciates your inclusion of that

statement, but the totaling the number of tests is not necessary for the paper.

Here we do not agree. We think to give the number of tests is crucial when indicating that no adverse effect occurred.

Table 2: Would it be possible to add a column of both males and females averaged, in the results

you perform analyses on these groups combined, therefore it would aid in interpretation.

Thank you for the recommendation. We now have added a corresponding column.

Table 3: This may just be an uploading/downloading issue, but in the exercise column, “[kg]”

should be moved up to sit next to the exercise name.

Indeed this is a up-/downloading issue due to the submission template of mdpi. I will also be aware in the proofs, that “kg” will be placed next to the exercise name.

Line 166 – 182: This section should be reorganized. IT is difficult to read the equation and

understand what when into it. I suggest the addition of an equation figure with reorganization of

what’s to be included. Likewise for Lines: 195-210.

We slightly restructured the paragraph. Now ewe feel it is very comprehensible and clearly explained our approach. Additionally, we now have added the equation figure.

Line 195: It is not clear what the Isometric co-efficient is or how this was calculated.

Thank you for the recommendation. We now have added information on calculation of the Isometric co-efficient  (Iso-coeff: Mean of 1RPM/ISOM).

Discussion

Line 233-238: This section seems to be contradictory. It reads that in this population exercise

specific prediction equations are superior, but the difference between exercise specific and leg-

press based prediction equations are negligible. This would suggest that the adoption of exercise

specific prediction equations is arbitrary.

We now have slightly revised the sentence to increase clarity and comprehensibility. First of all our result indicates that the cohort and exercise specific prediction equation provided the most accurate performance. We observed significantly more accurate prognostic performance compared with available equations. This, however, only partly refers to the KLW-equation that addressed a roughly comparable cohort of trained postmenopausal women.

Nevertheless, unexpectedly, the comparison of the prediction equation that based on the leg-press exercise and the exercise specific equation did not lead to significant differences in favor of the exercise specific prediction equation. Correspondingly, yes, this suggests at least that among our seven devices and for this cohort, the relevance of exercise specific prediction equations is not crucial, nevertheless, will slightly increase the prediction performance for most but not mandatory all exercises.

Line 248: Do you think this is a coincidence or does the diagnosis of osteopenia/osteoporosis in

the spine and femoral head play a role in leg-press and trunk flexion exercises not being

accurately predicted?

This is an interesting aspect. However, it is rather unlikely that  there is a relationship between low BMD and corresponding site specific muscular performance in this functionally fit cohort.  In addition, this cohort of post-menopausal women also suffers from osteopenia/osteoporosis.

Line 262: What is meant by “restricted to periods of intensive RT”?

This means “periods with low number of repetitions and repetition to fatigue”. We now have added this explanation.

Line 269: Why not look at rep ranges in the 2-4 RTF, from the evidence laid out, predictions get

more accurate the closer you get to 1RM. It would also be more specific to the energy systems

used during 1RM testing. Of course, the relevance of 1RM testing is determined by the specific

goals of each person.

Here we refer to our study and we did not test this RTF-range. However, from a pragmatic point of view, 2 to 4 reps to failure are a range rarely prescribed in exercise protocols for older adults, be it with or without osteopenia/osteoporosis.  

Line 289-295: The reproducibility of rep max testing is likely the most crucial aspect of this

study. In this section, it is not totally clear if you mean the 1RM CVs from this data set match

cohorts from another study (this is what I think you mean) or if the 1RM CVs you used were data

from another data set.

Thank you for this question. We agree that reproducibility of rep max testing but also for the RTF tests are very important for the validity of our prediction equation. We now have specified our proceeding for clarity.

Also, could you further explain the reproducibility data from RTF tests, where they differ by +/-

Thank you for this recommendation. We now have added a paragraph in the result section and added information to the corresponding paragraph in the method section.

  1. Did participants perform an additional set if they only completed 4 reps? From your methods,

I understand that by performing 9 reps on the 5-8RTF session, that data instead went to the 9-

12RTF rep range and the 5-8RTF was trialed again.

Yes, both is true.

In accordance with my previous comment here, do you think giving participants the ability to re-

do the 5-8RTF aided in the increased accuracy?

No, we do not expect that. At least the assessments of test- retest (intra-rater) reliability performed two-four weeks prior to the 1RM assessments did not indicate increased accuracy of the second test.

Line 299-300: “Considering further the present evidence that reproducibility and validity of

1RM testing are not dependent on RT experience [8, 29], transferring application of these RTF

equations to non-athletic cohorts seems appropriate.”

This reviewer suggests being careful of the language used here. To say that the validity and

reproducibility of 1RM testing are not dependent on RT experience is likely too broad. This

stamen may hold true here, but could quickly become false (e.g., different exercises, different

age groups, etc.).

We have referred to the meta-analysis of Grgic (Grgic et al., 2020). However, we agree, that this conclusion might be too far going. We now have revised the sentence.

References

Grgic, J., Lazinica, B., Schoenfeld, B. J., and Pedisic, Z. (2020). Test-Retest Reliability of the One-Repetition Maximum (1RM) Strength Assessment: a Systematic Review. Sports Med Open. 6: 31. doi: 10.1186/s40798-020-00260-z

Hettchen, M., von Stengel, S., Kohl, M., Murphy, M. H., Shojaa, M., Ghasemikaram, M., et al. (2021). Changes in Menopausal Risk Factors in Early Postmenopausal Osteopenic Women After 13 Months of High-Intensity Exercise: The Randomized Controlled ACTLIFE-RCT. Clin Interv Aging. 16: 83-96. doi: 10.2147/CIA.S283177

Kemmler, W., Bebenek, M., Kohl, M., and Von Stengel, S. (2015). Exercise and fractures in postmenopausal women. Final results of the controlled Erlangen Fitness and Osteoporosis Prevention Study (EFOPS). Osteoporos Int. 26: 2491-2499. doi: 10.1007/s00198-015-3165-3

Kemmler, W., Kohl, M., Fröhlich, M., Jakob, F., Engelke, K., von Stengel, S., et al. (2020). Effects of High Intensity Resistance Training on Osteopenia and Sarcopenia parameters in Older Men with Osteosarcopenia - One-year results of the randomized controlled Franconian Osteopenia and Sarcopenia Trial (FrOST). J Bone Miner Res. 35: 1634-1644. doi: 10.1002/jbmr.4027

Kemmler, W. K., Lauber, D., Wassermann, A., and Mayhew, J. L. (2006). Predicting maximal strength in trained postmenopausal woman. J Strength Cond Res. 20: 838-842. doi: R-18905 [pii]

10.1519/R-18905.1

Round 2

Reviewer 2 Report

Comments and Suggestions for Authors

See attached Document
